# Novel Pesticidal Efficacy of *Araucaria heterophylla* and *Commiphora molmol* Extracts against Camel and Cattle Blood-Sucking Ectoparasites

**DOI:** 10.3390/plants11131682

**Published:** 2022-06-24

**Authors:** Mohamed M. Baz, Hanem F. Khater, Rowida S. Baeshen, Abdelfattah Selim, Emad S. Shaheen, Yasser A. El-Sayed, Salama A. Salama, Maysa M. Hegazy

**Affiliations:** 1Department of Entomology, Faculty of Science, Benha University, Benha 13518, Egypt; yasser.abdelrahman@fsc.bu.edu.eg (Y.A.E.-S.); maysa.hegazy@fsc.bu.edu.eg (M.M.H.); 2Parasitology Department, Faculty of Veterinary Medicine, Benha University, Toukh 13736, Egypt; hanem.salem@fvtm.bu.edu.eg; 3Department of Biology, Faculty of Science, University of Tabuk, Tabuk 71421, Saudi Arabia; rbaeshen@ut.edu.sa; 4Department of Animal Medicine (Infectious Diseases), Faculty of Veterinary Medicine, Benha University, Toukh 13736, Egypt; abdelfattah.selim@fvtm.bu.edu.eg; 5Medical Research Centre, Jazan University, Jazan 45142, Saudi Arabia; emadshaheen@hotmail.com; 6Department of Biology, Faculty of Science, Jazan University, Jazan 45142, Saudi Arabia; sasalama@jazanu.edu.sa; 7Department of Zoology, Faculty of Science, Damanhour University, Damanhour 22511, Egypt

**Keywords:** *Boophilus annulatus*, *Hyalomma dromedarii*, *Hippobosca maculata*, *Haematopinus eurysternus*, phenols, sesquiterpene

## Abstract

Botanical insecticides are promising pest control agents. This research investigated the novel pesticidal efficacy of *Araucaria heterophylla* and *Commiphora molmol* extracts against four ectoparasites through treated envelopes. Seven days post-treatment (PT) with 25 mg/mL of *C. molmol* and *A. heterophylla*, complete mortality of the camel tick, *Hyalomma dromedarii* and cattle tick, *Rhipicephalus (Boophilus) annulatus* were reached. Against *H. dromedarii*, the median lethal concentrations (LC_50s_) of the methanol extracts were 1.13 and 1.04 mg/mL and those of the hexane extracts were 1.47 and 1.38 mg/mL, respectively. The LC_50_ values of methanol and hexane extracts against *R. annulatus* were 1.09 and 1.41 plus 1.55 and 1.08 mg/mL, respectively. Seven days PT with 12.5 mg/mL, extracts completely controlled *Haematopinus eurysternus* and *Hippobosca maculata*; LC_50_ of *Ha. eurysternus* were 0.56 and 0.62 mg/mL for methanol extracts and 0.55 and 1.00 mg/mL for hexane extracts, respectively, whereas those of *Hi. maculata* were 0.67 and 0.78 mg/mL for methanol extract and 0.68 and 0.32 mg/mL, respectively, for hexane extracts. *C. molmol* extracts contained sesquiterpene, fatty acid esters and phenols, whereas those of *A. heterophylla* possessed monoterpene, sesquiterpene, terpene alcohols, fatty acid, and phenols. Consequently, methanol extracts of *C. molmol* and *A. heterophylla* were recommended as ecofriendly pesticides.

## 1. Introduction

Blood-feeding arthropods are serious pests of worldwide distribution, including the camel tick, *Hyalomma dromedarii* (Koch, 1844); cattle tick, *Rhipicephalus (Boophilus) annulatus*, formerly *Boophilus annulatus* (Say, 1821), (Acari: Ixodidae); the adult cattle louse fly, *Hippobosca maculata* Leach (Diptera: Hippoboscidae); and the shortnosed cattle louse, *Haematopinus eurysternus* (Nitzsch, 1818), (Psocodea: Haematopinidae). Haematophagous pests cause dermal damage to be grazing animals, leading to severe economic loss because of blood loss, irritation, general stress, damaged skin and hide, retarded growth, weight loss, depression of the immune system, decreased meat and milk production, and transmission of life-threatening diseases [1,2,3].

The prevention of arthropod-borne diseases relies on effective pest management strategies [4,5,6]. Even though the employment of conventional pesticides and repellents represent a worthy solution to avoid arthropod bites, they resulted in serious environmental risks and unfavorable effects on non-target creatures, animals, and humans, and contaminated dairy and meat products [6] and development of resistant strains of pests; therefore, searching for alternative ways of pests control is an urgent need [3,7,8,9,10,11,12,13,14,15].

Some other approaches could be used for controlling pests, such as botanicals and biological control, vaccination, photopesticides, and acids [16,17,18,19,20,21,22,23]. Searching for alternative control strategies, mainly from plant-based resources, is a promising field [5].

Botanicals have been well- known for their medicinal properties [24] since ancient times [25] and induce anthelmintic, antiprotozoal, antiviral, antifungal, and antibacterial [26,27,28,29,30,31,32] and pesticidal effects [14,15] such as ovicidal [33,34], larvicidal and insect growth regulating effects [19,35,36,37,38,39,40,41,42,43,44,45,46,47,48] as well as adulticidal and repellent properties [8,33,34,39,45,46,49,50,51,52,53]. Botanicals are characterized by high efficiency against pests and prevention of their associated diseases, safety to non-target organisms [5,10,44], and biodegradation [5,11].

Myrrh oil-resin, *Commiphora molmol* Engler (Sapindales: Burseraceae) is an oleo-gum resin that grows in North-east Africa and was used as a house fumigant for pest control by Ancient Egyptians [25]. It has antiparasitic [54] and molluscicidal effects [25,55] and its sesquiterpene-rich fractions induce antibacterial and antifungal activities [56]. *C. molmol* has pesticidal effects against the green bottle fly and mosquitoes [57,58,59].

The Polynesian pine, *Araucaria heterophylla* Salisb (*A. excelsa*) (Pinales: Araucariaceae) is an ornamental evergreen coniferous tree. Araucaria plants exhibit several pharmaceutical potentials, including anti-inflammatory, antiulcerative, antiviral, antimicrobial, neuroprotective, and anti-depressant [60]. *A. heterophylla* has an insecticidal effect against mosquitoes [61,62]. It is worth mentioning that the safety of *C. molmol* [63,64] and *A. heterophylla* [59] had been confirmed. Because botanicals decay faster than most synthetic pesticides, they are more environmentally friendly and less likely to kill beneficial insects [14,15]. As a result, we hypothesize that *A. heterophylla* and *C. molmol* plant resins contain a variety of active biological components that could be used to control pests without contaminating the environment, making them a viable alternative to industrial pesticides. The study’s main goals were to investigate the novel pesticidal effect of methanol and hexane extracts of myrrh and Polynesian pine against four camel and cattle blood-sucking ectoparasites, calculate their lethal concentration values to kill 50, 90, and 95% of the exposed ectoparasites (LC_50_, _60_, and _95_, respectively), and investigated their phytochemical analyses.

## 2. Results and Discussion

### 2.1. Effect of the Plant Resin Extracts on Arthropods

Bloodsucking arthropods have an elegant method of delivery for a wide range of infectious agents [4], and their safe control is very crucial. This work evaluated two plant extracts of *A. heterophylla* and *C. molmol* against four arthropods, *H. dromedarii* (camel tick), *R. annulatus* (cattle tick), *Hi. maculata* (cattle louse fly), and *Ha. eurysternus* (cattle louse). The data expressed dose and time-dependent efficacy, a similar response was observed [52,65].

All plant extracts in this study showed moderate to high toxic effects against cattle and camel ectoparasites after 24 h of exposure, and methanol extracts were more effective than hexane extracts. The mortality percent (MO%) seven days PT of *H. dromedarii* with 12.5 mg/mL methanol extracts of *C. molmol* and *A. heterophylla* were 100% with LC_50_ (50%, median lethal concentration) = 1.13 and 1.04 mg/mL, respectively); whereas those of hexane extracts were 100% PT with 25 mg/mL (LC_50_ = 1.47 and 1.38 mg/mL, respectively (Table 1 and Table 2).

Similar to the response of camel ticks, the results of this work showed that plant extracts effectively controlled the cattle tick, *R. annulatus* because 100% mortality% was reached seven days PT with 12.5 mg/mL methanol extracts of *C. molmol* and *A. heterophylla* (LC_50_ = 1. 09 and 1.41 mg/mL, respectively) whereas those of hexane extracts were reached PT with 25 mg/mL (LC_50_ = 1.55 and 1.08%, respectively) (Table 3 and Table 4).

Analogous to our study, *Commiphora* spp. has an acaricidal effect, as *C. molmol* extract effectively controlled the fowl tick, *Argas persicus*, and its mortalities reached 63, 67, 76, 87, and 94% PT for 12 days PT with 0.625, 1.25, 2.5, 5, and 10%, respectively (LC_50_ = 1.28, 0.88, 0.84, 0.50, and 0.42% PT for 1, 2, 3, 6, and 12 days, respectively) [65].

*Commiphora swynnertonii* (Burtt) exudate had a parallel strong acaricidal effect against ticks such as *Rhipicephalus appendiculatus* and *Amblyioma variegatum* (LC_50_ = 1.72 and 1.91 mg/mL, respectively, and LC_99_ were 3.5 and 3.7 mg/mL, respectively) and adversely affected their reproduction capability [66]. *C. swynnertonii* (Burtt) stem bark exudate also induced an acaricidal effect against *Rhipicephalus appendiculatus* and exhibited a significant (*p* < 0.05) mortality and inhibition of laid eggs of ticks PT with concentrations over 25 and 90 mg/mL, respectively, and no hatching of eggs was observed in all treated groups [67]. A similar study revealed the adulticidal effect of the *C. swynnertonii* stem bark ethyl acetate, petroleum ether, and methanolic extracts against *R. appendiculatus* and *A. variegatum*. The petroleum ether extract exhibited higher acaricidal activity (LC_50_ = 72.31 and 71.67 mg/mL, respectively) and its MO%, 156 h PT, were 100 and 87% against *Amblyomma variegatum* and *Rhipicephalus appendiculatus*, respectively [67].

The gum Haggar, *Commiphora holtziana*, resin repelled the cattle tick, *Boophilus microplus* for up to 5 h with the hexane extract [68]. Additionally, myrrh not only controls ticks but also inhibited the propagation of blood parasites transmitted by ticks as bovine (*Babesia bovis, B. bigemina*, and *B. divergens*) and *equine piroplasms* (*Theileria equi* and *B. caballi*) [54]. *C. molmol* also induced molluscicidal and biological activities against *Biomphalaria alexandrina* and *Bulinus truncatus* (Mollusca: Gastropoda) [55].

Furthermore, the *C. molmol* resin extract displays pesticide action against many pests. It effectively controlled the blowfly, *Lucilia sericata* and its LC_50_ values were 6.03, 7.96, and 6.55 mg/mL for the first, second and third larva stages, respectively, and induced morphological abnormalities in larvae, pupae, and adults [58]. *C. molmol* was toxic to the fowl tick *Argas persicus* (LC_50_ = 1.28, 0.88, 0.84, 0.50 and 0.42 PT for one, two, three, six, and 12 days, respectively. Mortalities reached 63, 67, 76, 87, and 94% PT with 0.625, 1.25, 2.5, 5, and 10%, respectively [69].

Analogous studies showed the acaricidal effect of other plant extracts against ticks. Recently, the ethanol extracts of *Vitex castus* and *Zingiber officinale* had an acaricidal effect against *H. dromedarii*, as the mortality 15 days PT reached 80.8 and 84.7%, respectively, and LC_50_ values three days PT were 12.2 and 11.8%, respectively, whereas their median lethal time (LT_50_) values PT was 2.6 and 2.5 days, respectively [52]. Moreover, *Protium spruceanum* on resistant strains against *R. annulatus* induced mortality > 80 and 90% PT with 100 and 50 mg/mL ethanolic extract and ethyl acetate extracts, respectively [70]; ethyl alcohol and petroleum ether extracts of *Melia azedarach* and *Artemisia herba-alba* were also effective acaricides against embryonated eggs and engorged nymphs of *H. dromedarii* when compared to Butox^®^5.0 (Deltamethrin) [71].

A related study showed that the methanol extract of neem and *Citrullus colocynthis* produced an acaricidal effect against adult females, eggs, and larvae, and neem was more effective against *H. dromedarii* [72]. Some other materials are also effective in vitro acaricides such as peracetic acid against *Boophilus annulatus* and the fowl tick, *Argas persicus* [17] and *A. persicus*, infesting laying hens [18]. Moreover, some photosensitizers such as safranin and rose bengal had a strong acaricidal effect against *H. dromedarii* and suppressed the reproductive potential of its engorged females [16].

Lice infestation in cattle is mainly controlled by conventional insecticides [73], and to the best of our knowledge, there are no natural treatments for controlling such pests as. Data from this work showed that the methanol extracts of *C. molmol* and *A. heterophylla* effectively controlled the cattle lice, *Ha. eurysternus*, reaching 100% mortality PT with 6.35% of methanol extracts (LC_50_ = 0.56 and 0.62 mg/mL, respectively, and 96.67 and 83.33%, respectively, PT with 6.3% hexane extracts (LC_50_ = 0.55 and 1.00 mg/mL, respectively) (Table 5 and Table 6).

Studies about using botanicals against lice infesting large animals are very rare. A comparable study indicated that essential oils had in vitro and in vivo lousicidal potential against the buffalo louse, *Haematopinus tuberculatus* (Burmeister, 1839), in Egypt. Through filter paper contact bioassays, the LC_50_ values, four minutes PT, were 2.74, 12.35, 7.28, 22.79, and 18.67% for camphor (*Cinnamomum camphora*, Laurales: Lauraceae), peppermint (*Mentha piperita* L., Lamiales: Lamiaceae), onion (*Allium cepa*, Asparagales: Amaryllidaceae), rosemary oils (*Rosmarinus officinalis* Linn, Lamiales: Lamiaceae), and chamomile (*Matricaria chamomilla* L., Asterales: Asteracea), respectively, and oils induced ovicidal effects except rosemary, which was not applied [33]. Moreover, essential oils of garlic, clove, pumpkin, onion, and marjoram effectively controlled the dog louse, *Trichodectes canis* in vitro [51] and camphor oil controlled the slender pigeon louse, *Columbicola columbae*, in vitro and in vivo [49].

This investigation indicated that complete mortalities were reached seven days PT for the cattle louse fly, *Hi. maculata*, with 12.5 mg/mL extracts of *C. molmol* and *A. heterophylla* (LC_50_ values PT with methanol extract were 0.67 and 0.78 mg/mL, respectively, whereas those of hexane extracts were 0.68 and 0.32 mg/mL, respectively. After treatment with a lower concentration, 6.3%, MO% reached 100 and 93.33% PT with methanol extracts and 90 and 100% PT with hexane extracts (Table 7 and Table 8).

Parallel studies of using botanicals against *Hi. maculata* were also recorded. The leaf of *Ricinus communis*, *Malabarica malabarica*, and *Gloriosa superba* (methanol, chloroform, and chloroform extracts, respectively) effectively controlled *Hi. maculata* and the tick *Haemaphysalis bispinosa* [74]. The aqueous crude leaf extracts of Catharanthus roseus had insecticidal efficacy against the adults of *Hi. maculata* and the sheep-biting louse, *Bovicola ovis* (LD_50_ = 36.17 and 30.35 mg/L, respectively) [75].

A similar study proved the adulticidal activity of *Cissus quadrangularis* through an aqueous extract, AgNO_3_ solution, and synthesized Ag NPs against the cattle tick, *Rhipicephalus* (*Boophilus*) *microplus* larvae (LC_50_ = 50.00, 21.72, and 7.61 mg/L, respectively) and the adult of *Hi. maculata* (LC_50_ = 37.08, 40.35 and 6.30 mg/L, respectively) via the contact toxicity method [76].

Moreover, essential oils had repellent, adulticidal, larvicidal, and ovicidal effects against cycloraphan flies [34,38,39,42,53,77]. Essential oils and d-phenothrin repelled biting and non-biting flies infesting water buffalo, *Hippobosca equine*, *Haematobia irritans*, *Musca domestica*, and *Stomoxys calcitrans*, for six and three days PT, respectively [33].

It is worth mentioning that the essential oil of *Commiphora erythraea* (Opoponax) induced a larvicidal effect against *Culex restuans* Theobald, *Culex pipiens* L., and *Aedes aegypti* L. (LC_50_ = 19.05, 22.61, and 29.83 ppm, respectively) [57].

Likewise, in our findings, some Oil-resins had larvicidal activity against *Culex pipiens* such as *C. molmol*, *A. heterophylla*, *Boswellia sacra*, *Pistacia lentiscus*, and *Eucalyptus camaldulensis*. After treatment for 24 and 48 h PT with 1500 ppm, the best effect was observed PT with acetone extracts of *C. molmol*, 83.3% and 100% with LC_50_ values were 623.52 and 300.63 ppm, as well as *A. heterophylla*, 75% and 95% with LC_50_ values, were 826.03 and 384.71 ppm, respectively. On the other hand, the aqueous extract of *A. heterophylla* was highly effective against *Cx. pipiens* (LC_50_ = 2819.85 and 1652.50 ppm) followed by *C. molmol* (LC_50_ = 3178.22 and 2322.53 ppm) 24 and 48 h PT, respectively [59]. As mosquito larvicides, *A. heterophylla* and *Azadirachta indica* (gum polysaccharides) were used for encapsulation of cyfluthrin-loaded superparamagnetic iron oxide nanoparticles [61].

### 2.2. Biochemical Analysis

It was noticed that most of the compounds belong to sesquiterpene, fatty acid esters and phenols were the most common compounds found in the methanol and hexane extracts of the myrrh, *C. molmol* plant while monoterpene, sesquiterpene, terpene alcohols, fatty acid, and phenols were found in in methanol and hexane extracts of *A. heterophylla* plant in larger amount.

Phytochemical analysis of this work revealed that the constituents of *C. molmol* and *A. heterophylla* extracts were identified by GC–MS analysis (Table 9, Table 10, Table 11 and Table 12) indicating that *C. molmol* and *A. heterophylla* contained the main chemical compounds 1,8,11,14-Heptadecatetraene, (Z,Z,Z)-(16.27%), 2(3H)-Benzofuranone, 6-ethenylhexahydro-6-methyl-3-methylene-7-(1-methylethenyl)-, [3aS-(3aà,6à,7á,7aá)]-(22.67%), Azuleno [4,5-b]furan-2(3H)-one, decahydro-3,6,9-tris(methylene)-, [3aS-(3aà,6aà,9aà,9bá)]-(47.28) and 1,8,11,14-Heptadecatetraene, (Z,Z,Z)-(7.43), 2(3H)-Benzofuranone, 6-ethenylhexahydro-6-methyl-3-methylene-7-(1-methylethenyl)-, [3aS-(3aà,6à,7á,7aá)]-(19.90), and ETHANONE, 1-(7,8-DIHYDRO-3-HYDROXY-4-PROPYL-2-NAPHTHALENYL)-(67.27%) for methanol and hexane extracts.

*C. molmol* methanol extract in the present study mainly contained benzofuran, 6-ethenyl-4,5,6,7-tetrahydro-3,6-dimethyl-5-isopropenyl-, trans-(15.35%), 1-NAPHTHALENOL, 4,7-DIMETHYL-2-(1-METHYLETHYL)-(13.80%), (R,5E,9E)-8-Methoxy-3,6,10-trimethyl-4,7,8,11-tetrahydrocyclodeca[b]furan (12.72%), and 6-[1-(HYDROXYMETHYL)VINYL]-4,8A-DIMETHYL-3-OXO-1,2,3,5,6,7,8,8A-OCTAHYDRO-2-NAPHTHALENYL ACETATE (10.35). On the other hand, *C. molmol* hexane extract mainly contained benzofuran, 6-ethenyl-4,5,6,7-tetrahydro-3,6-dimethyl-5-isopropenyl-, trans-(12.09%), NAPHTHALENE, 4-METHOXY-1,2,6,8-TETRAMETHYL-931.98%), (4aS,8aS)-3,8a-Dimethyl-5-methylene-4,4a,5,6,8a,9-hexahydronaphtho [2,3-b]furan (8.15%).

The chemical analysis in this study indicated that *A. heterophylla* contains the main chemical compounds the à-Pinene (3.24%), CYCLOHEXENE, 1-METHYL-4-(1-METHYLETHENYL)-(12.95%), 6-Tridecene, (Z)-99.34%), Copaene (7.96%), and Caryophyllene oxide (10.39%) for methanol extract and AZETIDINE-D1 (8.28%), 9-OCTADECENOIC ACID (Z)-(14.60%), Hexadecanoic acid, ethyl ester (9.57%), and CHOLEST-5-EN-3-OL (3á)-(19.15%) for hexane extract.

Parallel studies demonstrated that the Araucariaceae family including *A. heterophylla*, produces several monoterpenes, such as pinene, camphene, and limonene as common compounds [78]. *Araucaria* spp. contains various sesquiterpenes like humulanes, cadinanes, caryophyllanes, and other compounds [79]. The resin of *Araucaria columnaris* is rich in aromadendrene and bicyclogermacrene and contains sesquiterpene hydrocarbons and oxygenated sesquiterpenes [80,81]. Similar studies indicated that *A. heterophylla* contained flavonoids, sesqui and di-terpenes, and phenylpropanoids [81]; two monoterpene resins, b-pinene and a-pinene, were commonly found in wood found in *Araucaria angustifolia* and such compounds were detected in Norway spruce with many monoterpenoids in wood and bark [82].

Similar to our findings, GC–MS analysis revealed the presence of 4,4’-Dimethyl-2,2’- dimethylenebicyclohexyl-3,3’-diene (14.62%) and Copaene (13.64%) as the most prevailing constituents in *C. molmol* and *A. heterophylla*, respectively [59]. Bisabolene was the most abundant component in *Commiphora erythraea* essential oil (33.9%), fraction 2 (62.5%), and fraction 4 (23.8%), curzerene (32.6%), and α-santalene (30.1%) were the dominant chemical constituents in fractions 1 and 3, respectively [57]. Similar studies indicated that two resins, *Commiphora myrrha* and *Commiphora africana*, are rich in sesquiterpenes and sesquiterpene lactones through GC-MS analysis with anti-inflammatory and anticancer potential [83].

Finally, our data and others confirm that the presence of many secondary metabolites such as sesquiterpenes, phenols, aromatic terpenoids, fatty alcohol, eugenol, and many other bio-effective compounds may explain the effectiveness of *A. heterophylla* and *C. molmol* resin extracts against insect pests [82,84,85].

Phenolics are linked to toxicity against because they are important in plant-herbivore and pathogen interactions. Antioxidant characteristics were found in phenolic chemicals, which are thought to be the primary cause of the pesticide effect in nature [86]. The mode of action of *C. molmol* extract was revealed through histopathological and transmission election microscope of treated *A. persicus* via penetrating the cuticle towards the body cavity of treated ticks, destroying the epithelial gut cells, and ultimately resulted in the death of ticks. Moreover, lysing of epithelial gut cells with an irregularly distributed nucleus was commonly PT with low concentrations and rarely PT with high concentrations of *C. molmol*, whereas lysed epithelial gut cells (without nucleus or with aggregated one beside the basal lamina) were commonly observed PT with high concentrations and rare recorded PT with low concentrations [65,69]. Using plant-based pesticides had minimum or low toxicity for non-target organisms [5]. Specifically, the safety of *Commiphora* spp. was confirmed after oral toxicity in mice and rats [63].

## 3. Materials and Methods

### 3.1. Pest Collections

The collection of the adult stage of four pests of mixed sex was done from May to July 2021. The camel tick, *Hyalomma dromedarii* (Koch, 1844) and cattle tick, *Rhipicephalus (Boophilus) annulatus*, formerly *Boophilus annulatus* (Say, 1821), (Acari: Ixodidae), were collected from areas around infested camel and cattle, respectively, at the slaughterhouse in Jazan Province, Saudi Arabia. The adult cattle louse fly, *Hippobosca maculata* Leach (Diptera: Hippoboscidae) was collected from infested cattle mainly in the ears and tails. The cattle louse, *Haematopinus eurysternus*, was collected from the dewlap, cheeks, neck, flank, withers, and back of infested cattle. Pests were collected from and around animals that had no previous exposure to pesticides.

### 3.2. Collection of Plant Materials

*A. heterophylla* and *C. molmol* were collected from different areas in Saint Catherine (28°33′42″ N, 33°56′57″ E, altitude 2624), South Sinai Governorate, Egypt in May 2021. *C. molmol* resin was obtained as amber solid crystals, while *A. heterophylla* resin was a flexible white colloidal form (Figure 1). Plants were identified at the Flora and Phytotaxonomic section of the Agricultural Research Center in Giza, Egypt.

### 3.3. Preparation of Plant Extracts

Stock solutions of the plant oil-resins *A. heterophylla* and *C. molmol* were extracted by mechanically grinding 50 g of both plant oil-resins using a stainless-steel electric mixer and placing the powder in a Soxhlet apparatus for 6–8 h according to the type of solvent. Methanol and hexane were used as solvent, individually. The solution was filtered using Whatman No. 1 filter paper through a Buchner funnel, and the extracts were dried in an oven at 30 °C for 6 h. The extracts were stored in a dark bottle in a refrigerator at −5 °C for 24 h prior to the experiment [52].

### 3.4. Bioassays

The pesticide effectiveness of methanol and hexane extracts of *A. heterophylla* and *C. molmol* was evaluated against four ectoparasites, *H. dromedarii*, *R. annulatus*, *Hi. maculate*, and *Ha. eurysternus*. Preliminary experiments each containing 30 adult pests, grouped in three replicates, were made to evaluate the range of concentrations used for each pest. Treated envelopes were used [74]. The adult cattle and camel ticks were treated with the following concentrations: 1.6, 3.1, 6.3, 12.5, 25 mg/mL, while adult cattle louse fly and cattle louse were treated with the following concentrations: 0.8, 1.6, 3.1, 6.3, 12.5 mg/mL. Three replicates (each contained ten adult pests) were used for each concentration.

Each group of pests were added to a filter paper envelope, Whatman filter paper No.1, 125 mm diameter, and treated with a single concentration of the plant extracts as 3 mL test solution uniformly distributed with a pipette on internal surfaces of the envelopes. The control envelopes were impregnated with distilled water. The opening of the envelopes was folded and secured with a metallic clip with its identification marks like tested solution and concentration. Each treated replicate of pests was transported to a Petri dish lined with a filter paper. Treated pests were kept at 28 ± 2 °C and a relative humidity of 80 ± 5%. Mortalities were recorded one, three and seven days post-treatment (PT).

### 3.5. Biochemical Analysis

Biochemical analyses were made using GC/MS, a Thermo Scientific Trace GC Ultra/ISQ Single Quadrupole MS, TG-5MS fused silica capillary column, 0.1 mm, 0.251 mm, and 30 m thick. An electronic ionizer with 70 eV ionization energy was used. Helium gas was utilized as a carrier gas (flow rate = 1 mL/min). The injector and MS transmission line were set at 280 °C. The oven temperature was set at 50 °C, then increased to 150 °C at a rate of 7 °C per minute, then to 270 °C at a rate of 5 °C per minute (wait for 2 min), and finally to 310 °C at a rate of 3.5 °C/min (continued for 10 min). To investigate the quantification of all components found, a relative peak area was used. By comparing the retention periods and mass spectra of the chemicals with those of NIST, Willy Library data from the GC-MS instrument, and the chemicals were tentatively identified. The collective spectra of user-generated reference libraries were used for identification. Single-ion chromatographic reconstructions were used to assess peak homogeneity. Co-chromatographic analysis of reference compounds was performed whenever possible to confirm GC retention times [87,88].

### 3.6. Data Analyses

The data were analyzed by the software, SPSS V23 (IBM, New York, NY, USA), for doing the Probit analyses to calculate the lethal concentration (LC) values and the one-way analysis of variance (ANOVA) (Post Hoc/Turkey’s HSD test). The significant levels were set at *p* < 0.05.

## 4. Conclusions

It is crucial to safeguard livestock and domesticate animals from blood-feeding ectoparasites and vector-borne diseases. Worldwide, pest control is dependent on conventional pesticides, but resistance has developed to almost all classes of pesticides. Botanicals as eco-friendly pesticides represent conspicuous alternatives because of the wide diversity and high effectiveness of several plant-borne compounds. This study revealed, for the first time according to our knowledge, the efficacy of methanol and hexane extracts of *C. molmol* and *A. heterophylla* against four camel and cattle blood-sucking arthropods.

Our results confirmed that cattle lice and the louse fly were more susceptible (12.5 mg/mL) than cattle and camel ticks (25.0 mg/mL) to *A. heterophylla* and *C. molmol* extracts. Both methanol extracts were recommended as an ideal eco-friendly and inexpensive pest control approach that could be incorporated into integrated pest management used for the protection of large animals from vectors and vector-borne diseases. Further studies could be directed towards the field application and safety profile of *C. molmol* and *A. heterophylla* against non-target organisms as well as studying the synergistic effects of surfactants.

## Figures and Tables

**Figure 1 plants-11-01682-f001:**
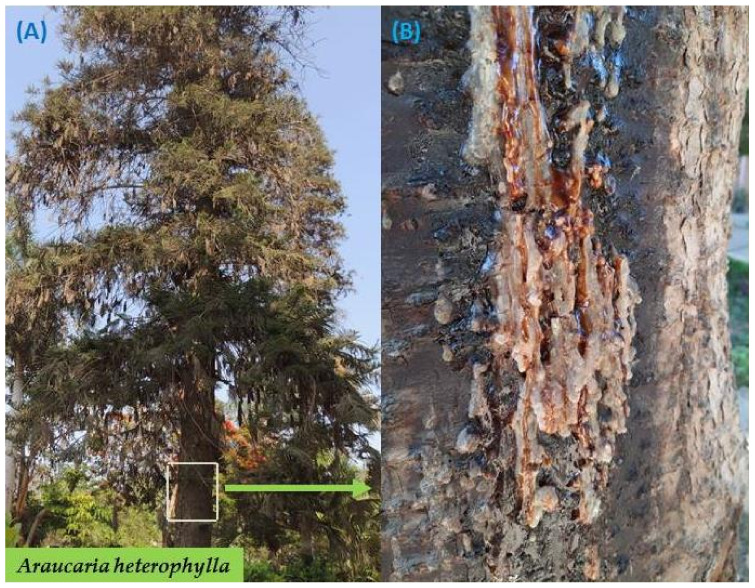
The tree of *Araucaria heterophylla* in Saint Catherine area (**A**), massive resinous sap outpouring of the trunk *Araucaria heterophylla* (**B**).

**Table 1 plants-11-01682-t001:** The efficacy of the plant extracts of *Commiphora molmol* and *Araucaria heterophylla* against the Camel tick, *Hyalomma dromedarii*.

		Mortality % (Mean ± SE)					
Plant Extracts	Concentration (mg/mL)	Methanol			Hexane		
		1 Day	3 Days	7 Days	1 Day	3 Days	7 Days
*Commiphora molmol*	0	0.00 ± 0.0 fC *	3.33 ± 3.33 fB	6.67 ± 3.33 eA	0.00 ± 0.0 fC	3.33 ± 3.33 fB	6.67 ± 3.33 fA
	1.6	16.67 ± 6.67 eC	36.67 ± 3.33 eB	63.33 ± 3.33 dA	13.33 ± 3.33 eC	30.00 ± 5.77 eB	53.33 ± 6.67 eA
	3.1	30.00 ± 5.77 dC	63.33 ± 3.33 dB	86.67 ± 3.33 cA	23.33 ± 3.33 dC	46.67 ± 3.33 dB	66.67 ± 6.67 dA
	6.3	53.33 ± 3.33 cC	73.33 ± 3.33 cB	93.33 ± 3.33 bA	40.00 ± 5.77 cC	66.67 ± 6.67 cB	76.67 ± 6.67 cA
	12.5	73.33 ± 8.82 bC	80.00 ± 5.77 bB	100.0 ± 0.00 aA	60.00 ± 5.77 bC	73.33 ± 8.82 bB	86.67 ± 8.82 bA
	25	86.67 ± 6.67 aC	90.00 ± 5.77 aB	100.0 ± 0.00 aA	76.67 ± 13.33 aC	80.00 ± 5.77 aB	100.0 ± 0.00 aA
*Araucaria heterophylla*	0	0.00 ± 0.0 fC	3.33 ± 3.33 fB	6.67 ± 3.33 dA	0.00 ± 0.0 fC	3.33 ± 3.33 fB	6.67 ± 3.33 fA
	1.6	20.00 ± 0.00 eC	46.67 ± 3.33 eB	73.33 ± 3.33 cA	13.33 ± 6.67 eC	33.33 ± 3.33 eB	56.67 ± 8.82 eA
	3.1	43.33 ± 8.82 dC	73.33 ± 3.33 dB	86.67 ± 8.82 bA	30.00 ± 5.77 dC	56.67 ± 3.33 dB	76.67 ± 8.82 dA
	6.3	63.33 ± 8.82 cC	80.00 ± 10.00 cB	100.0 ± 0.00 aA	43.33 ± 3.33 cC	66.67 ± 6.67 cB	80.00 ± 5.77 cA
	12.5	80.00 ± 5.77 bC	90.00 ± 5.77 bB	100.0 ± 0.00 aA	60.00 ± 5.77 bC	76.67 ± 3.33 bB	90.0 ± 10.00 bA
	25	93.33 ± 3.33 aC	100.0 ± 0.00 aB	100.0 ± 0.00 aA	76.67 ± 3.33 aC	90.00 ± 10.00 aB	100.0 ± 0.00 aA

* letters refer to significant difference; a–f: There is no significant difference (*p* > 0.05) between any two means, those within the same column have the same superscript letter; A, B & C: There is no significant difference (*p* > 0.05) between any two means for the same solvent, those within the same row have the same superscript letter. Three replicates were used for each concentration and 10 adult pests per replicate were used.

**Table 2 plants-11-01682-t002:** Lethal concentration values of plant extracts of *Commiphora molmol* and *Araucaria heterophylla* against *Hyalomma dromedarii*.

Days	Plant Extracts	Solvents	LC_50_ (95%CL) *	LC_90_ (95%CL)	LC_95_ (95%CL)	Equation **	X^2^
**1**	*Commiphora molmol*	Methanol	5.76 (4.91–6.75)	30.29 (22.91–44.26)	48.48 (34.50–77.54)	1.779 ± 0.157X	0.975
		Hexane	36.08 (31.12–56.66)	442.56 (315.42–498.16)	1731 (1420.15–2125.02)	0.613 ± 0.140X	40.179
	*Araucaria heterophylla*	Methanol	4.16 (3.52–4.85)	19.94 (15.69–27.49)	31.10 (23.15–46.50)	1.880 ± 0.165X	0.705
		Hexane	8.07 (6.71–9.86)	60.03 (40.13–107.74)	106.00 (64.94–217.67)	1.471 ± 0.149X	0.867
**3**	*Commiphora molmol*	Methanol	2.47 (1.78–3.15)	24.22 (17.05–41.08)	46.25 (29.28–93.94)	1.293 ± 0.153X	4.165
		Hexane	4.08 (3.12–5.11)	48.16 (30.61–97.54)	96.95 (54.49–241.38)	1.195 ± 0.144X	0.350
	*Araucaria heterophylla*	Methanol	1.78 (1.31–2.22)	9.85 (7.84–13.47)	15.99 (11.94–24.42)	1.726 ± 0.188X	6.546
		Hexane	3.17 (2.42–3.93)	28.26 (19.87–47.55)	52.53 (33.48–103.88)	1.349 ± 0.151X	2.469
**7**	*Commiphora molmol*	Methanol	1.13 (0.79–1.45)	4.55 (3.85–5.54)	6.76 (5.55–8.80)	2.121 ± 0.233X	1.488
		Hexane	1.47 (0.79–2.13)	24.60 (15.41–56.27)	54.62 (28.92–175.54)	1.049 ± 0.167X	0.793
	*Araucaria heterophylla*	Methanol	1.04 (0.67–1.33)	3.13 (2.65–3.92)	4.27 (3.49–5.95)	2.687 ± 0.433X	0.199
		Hexane	1.38 (0.90–1.84)	10.09 (7.80–14.53)	17.73 (12.65–29.71)	1.483 ± 0.182X	7.301

* LC_50_, _60_, and _95_ values = lethal concentration that kills 50, 90, and 95% of the exposed ectoparasite; (95%CL) = lower and upper confidence limit; ** Regression line equation; X^2^ = chi-square; Significant at *p* < 0.05 level.

**Table 3 plants-11-01682-t003:** Efficacy of the plant extracts *Commiphora molmol* and *Araucaria heterophylla* on Cattle ticks, *Rhipicephalus (Boophilus) annulatus*.

		Mortality % (Mean ± SE)					
Plant Extracts	Concentration (mg/mL)	Methanol			Hexane		
		1 Day	3 Days	7 Days	1 Day	3 Days	7 Days
*Commiphora molmol*	0	0.00 ± 0.0 fC *	3.33 ± 3.33 fB	6.67 ± 3.33 eA	0.0 ± 0.0 fC	3.33 ± 3.33 fB	6.67 ± 3.33 fA
	1.6	20.00 ± 5.77 eC	40.00 ± 0.00 eB	66.67 ± 3.33 dA	16.67 ± 3.33 eC	33.33 ± 6.67 eB	56.67 ± 8.82 eA
	3.1	33.33 ± 8.82 dC	70.00 ± 5.77 dB	90.00 ± 5.77 cA	26.67 ± 3.33 dC	50.00 ± 5.77 dB	70.00 ± 10.00 dA
	6.3	56.67 ± 3.33 cC	76.67 ± 3.33 cB	96.67 ± 3.33 bA	43.33 ± 6.67 cC	70.00 ± 5.77 cB	80.00 ± 5.77 cA
	12.5	70.00 ± 5.77 bC	83.33 ± 3.33 bB	100.0 ± 0.00 aA	56.67 ± 3.33 bC	76.67 ± 8.82 bB	90.00 ± 5.77 bA
	25	83.33 ± 3.33 aC	93.33 ± 6.67 aB	100.0 ± 0.00 aA	70.00 ± 5.77 aC	83.33 ± 3.33 aB	100.0 ± 0.00 aA
*Araucaria heterophylla*	0	0.00 ± 0.0 f C	3.33 ± 3.33 fB	6.67 ± 3.33 dA	0.0 ± 0.0 fC	3.33 ± 3.33 fB	6.67 ± 3.33 fA
	1.6	23.33 ± 3.33 eC	50.00 ± 5.77 eB	76.67 ± 6.67 cA	16.67 ± 8.82 eC	36.67 ± 3.33 eB	60 ± 10.00 eA
	3.1	46.67 ± 12.02 dC	76.67 ± 3.33 dB	86.67 ± 8.82 bA	40.00 ± 5.77 dC	60.00 ± 5.77 dB	80 ± 11.55 dA
	6.3	66.67 ± 12.02 cC	83.33 ± 12.02 cB	100.0 ± 0.00 aA	53.33 ± 8.82 cC	70.00 ± 10.00 cB	83.33 ± 8.82 cA
	12.5	83.33 ± 8.82 bC	93.33 ± 3.33 bB	100.0 ± 0.00 aA	63.33 ± 6.67 bC	80.00 ± 0.00 bB	93.33 ± 6.67 bA
	25	96.67 ± 3.33 aC	100.0 ± 0.00 aB	100.0 ± 0.00 aA	80.00 ± 5.77 aC	93.33 ± 6.67 aB	100.0 ± 0.00 aA

* letters refer to significant difference; a–f: There is no significant difference (*p* > 0.05) between any two means, those within the same column have the same superscript letter; A, B & C: There is no significant difference (*p* > 0.05) between any two means for the same solvent, those within the same row have the same superscript letter. Three replicates were used for each concentration and ten numbers of adult pests per replicate were used.

**Table 4 plants-11-01682-t004:** Lethal concentration values of plant extracts of *Commiphora molmol* and *Araucaria heterophylla* against *Rhipicephalus (Boophilus) annulatus*.

Days	Plant Extracts	Solvents	LC_50_ (95%CL) *	LC_90_ (95%CL)	LC_95_ (95%CL)	Equation **	X^2^
**1**	*Commiphora molmol*	Methanol	5.26 (4.62–6.65)	38.19 (27.18–61.90)	65.95 (43.34–120.72)	1.530 ± 0.150X	0.628
		Hexane	9.24 (7.46–11.85)	96.45 (56.88–218.26)	187.47 (98.46–512.05)	1.258 ± 0.145X	0.172
	*Araucaria heterophylla*	Methanol	3.68 (3.10–4.28)	17.30 (14.05–22.58)	26.81 (20.77–37.55)	1.908 ± 0.153X	0.568
		Hexane	6.09 (4.99–7.43)	51.83 (34.48–94.53)	95.07 (57.43–201.82)	1.379 ± 0.147X	3.420
**3**	*Commiphora molmol*	Methanol	2.42 (0.75–3.58)	16.91 (13.56–88.54)	29.31 (26.34–256.46)	1.520 ± 0.150X	10.917
		Hexane	3.40 (2.55–4.27)	37.60 (24.87–70.99)	74.29 (43.71–170.66)	1.228 ± 0.147X	2.769
	*Araucaria heterophylla*	Methanol	1.41 (0.82–2.25)	12.97 (8.12–17.10)	24.33 (18.12–32.14)	1.330 ± 0.200X	25.761
		Hexane	2.71 (2.06–3.37)	21.64 (15.81–34.10)	38.96 (26.03–71.10)	1.422 ± 0.156X	2.660
**7**	*Commiphora molmol*	Methanol	1.09 (0.70–1.40)	3.58 (2.98–4.72)	5.00 (3.96–7.56)	2.496 ± 0.423X	0.875
		Hexane	1.55 (1.07–2.02)	10.62 (8.26–15.16)	18.31 (13.17–30.02)	1.537 ± 0.181X	5.304
	*Araucaria heterophylla*	Methanol	1.41 (0.72–1.89)	12.97 (11.52–18.78)	24.33 (22.14–34.15)	1.330.6 ± 0.433X	25.76
		Hexane	1.08 (0.65– 1.86)	10.11 (8.44– 16.10)	19.03 (12.45– 27.10)	1.323±0.193X	11.720

* LC_50_, _60_, and _95_ values = lethal concentration that kills 50, 90, and 95% of the exposed ectoparasite; (95%CL) = lower and upper confidence limit; ** Regression line equation; X^2^ = chi-square; Significant at *p* < 0.05 level.

**Table 5 plants-11-01682-t005:** Efficacy of the plant extracts of *Commiphora molmol* and *Araucaria heterophylla* on cattle lice, *Haematopinus eurysternus*.

		Mortality % (Mean ± SE)					
Plant Extracts	Concentration (mg/mL)	Methanol			Hexane		
		1 Day	3 Days	7 Days	1 Day	3 Days	7 Days
*Commiphora molmol*	0	0.00 ± 0.0 fC *	3.33 ± 3.33 fB	6.67 ± 3.33 eA	0.0 ± 0.0 eC	3.33 ± 3.33 fB	6.67 ± 3.33 fA
	0.8	20.00 ± 5.77 eC	40.00 ± 0.00 eB	66.67 ± 3.33 dA	20.00 ± 0.00 dC	36.67 ± 8.82 eB	63.33 ± 3.33 eA
	1.6	33.33 ± 8.82 dC	70.00 ± 5.77 dB	90.00 ± 5.77 cA	30.00 ± 5.77 C	60.00 ± 0.00 dB	83.33 ± 3.33 dA
	3.1	56.67 ± 3.33 cC	76.67 ± 3.33 cB	96.67 ± 3.33 bA	50.00 ± 5.77 cC	76.67 ± 3.33 cB	90.00 ± 5.77 cA
	6.3	83.33 ± 3.33 bC	90.00 ± 5.77 bB	100.0 ± 0.00 aA	73.33 ± 6.67 bC	83.33 ± 3.33 bB	96.67 ± 3.33 bA
	12.5	100.0 ± 0.00 aC	100.0 ± 0.00 aB	100.0 ± 0.00 aA	86.67 ± 6.67 aC	93.33 ± 3.33 aB	100 ± 0.00 aA
*Araucaria heterophylla*	0	0.00 ± 0.0 fC	3.33 ± 3.33 fB	6.67 ± 3.33 eA	0.00 ± 0.0 fC	3.33 ± 3.33 fB	6.67 ± 3.33 fA
	0.8	16.67 ± 3.33 eC	33.33 ± 6.67 eB	63.33 ± 6.67 dA	13.33 ± 3.33 eC	30.00 ± 5.77 eB	53.33 ± 8.82 eA
	1.6	26.67 ± 3.33 dC	60.00 ± 5.77 dB	80.00 ± 5.77 cA	23.33 ± 8.82 dC	40.00 ± 10.00 dB	63.33 ± 3.33 dA
	3.1	46.67 ± 3.33 cC	70.00 ± 10.00 cB	83.33 ± 8.82 bA	36.67 ± 3.33 cC	53.33 ± 3.33 cB	70.00 ± 0.00 cA
	6.3	56.67 ± 3.33 bC	76.67 ± 3.33 bB	100.0 ± 0.00 aA	46.67 ± 6.67 bC	70.00 ± 10.00 bB	83.33 ± 6.67 bA
	12.5	76.67 ± 3.33 aC	90.00 ± 5.77 aB	100.0 ± 0.00 aA	63.33 ± 3.33 aC	76.67 ± 3.33 aB	100.0 ± 0.00 aA

* letters refer to significant difference; a–f: There is no significant difference (*p* > 0.05) between any two means, within the same column they have the same superscript letter; A, B & C: There is no significant difference (*p* > 0.05) between any two means for the same solvent, within the same row they have the same superscript letter. Three replicates were used for each concentration and ten numbers of adult pests per replicate were used.

**Table 6 plants-11-01682-t006:** Lethal concentration values of plant extracts of *Commiphora molmol* and *Araucaria heterophylla* against *Haematopinus eurysternus*.

Days	Plant Extracts	Solvents	LC_50_ (95%CL) *	LC_90_ (95%CL)	LC_95_ (95%CL)	Equation **	X^2^
**1**	*Commiphora molmol*	Methanol	2.27 (1.97–2.59)	8.49 (6.94–11.04)	12.34 (9.66–17.05)	2.240 ± 0.180	5.034
		Hexane	2.88 (2.43–3.39)	16.37 (12.16–24.66)	26.79 (18.61–44.58)	1.698 ± 0.561	1.345
	*Araucaria heterophylla*	Methanol	4.08 (3.36–5.04)	33.78 (21.83–64.49)	61.48 (36.10–136.45)	1.397 ± 0.148	1.329
		Hexane	6.65 (5.20–9.21)	80.76 (42.32–229.34)	163.87 (75.16–581.83)	1.183 ± 0.148	0.397
**3**	*Commiphora molmol*	Methanol	1.09 (086–1.32)	5.23 (4.22–6.99)	8.15 (6.21–11.93)	1.889 ± 0.190	6.631
		Hexane	1.28 (0.97–1.57)	8.91 (6.73–13.25)	15.45 (10.78–26.14)	1.521 ± 0.161	2.058
	*Araucaria heterophylla*	Methanol	1.47 (1.11–1.84)	13.48 (9.49–22.75)	25.25 (16.06–50.33)	1.334 ± 0.152	4.287
		Hexane	2.75 (2.14–3.47)	36.43 (21.46–84.85)	75.78 (39.01–221.81)	1.142 ± 0.143	0.662
**7**	*Commiphora molmol*	Methanol	0.56 (0.38–0.71)	1.77 (1.49–2.24)	2.44 (1.99–3.38)	2.589 ± 0.379	0.876
		Hexane	0.55 (0.35–0.73)	2.86 (2.32–3.80)	4.57 (3.49–6.85)	1.791 ± 0.234	1.514
	*Araucaria heterophylla*	Methanol	0.62 (0.24–1.12)	3.07 (2.24–4.15)	4.82 (2.68–6.20)	1.856 ± 0.229	11.223
		Hexane	1.00 (0.64–1.68)	8.37 (5.88–11.32)	15.27 (9.85–21.15)	1.392 ± 0.159	11.114

* LC_50_, _60_, and _95_ values = lethal concentration that kills 50, 90, and 95% of the exposed ectoparasite; (95%CL) = lower and upper confidence limit; ** Regression line equation; X^2^ = chi-square; Significant at *p* < 0.05 level.

**Table 7 plants-11-01682-t007:** Efficacy of the plant extracts of *Commiphora molmol* and *Araucaria heterophylla* against the cattle louse fly, *Hippobosca maculata*.

		Mortality % (Mean ± SE)					
Plant Extracts	Concentration (mg/mL)	Methanol			Hexane		
		1 Day	3 Days	7 Days	1 Day	3 Days	7 Days
*Commiphora molmol*	0	0.00 ± 0.0 fC *	3.33 ± 3.33 fB	6.67 ± 3.33 eA	0.00 ± 0.0 fC	3.33 ± 3.33 fB	6.67 ± 3.33 fA
	0.8	20.00 ± 5.77 eC	33.33 ± 6.67 eB	60.00 ± 5.77 dA	13.33 ± 6.67 eC	30.00 ± 5.77 eB	56.67 ± 3.33 eA
	1.6	40.00 ± 5.77 dC	60.00 ± 5.77 dB	83.33 ± 3.33 cA	23.33 ± 8.82 dC	53.33 ± 6.67 dB	76.67 ± 3.33 dA
	3.1	53.33 ± 3.33 cC	73.33 ± 3.33 cB	90.00 ± 5.77 bA	43.33 ± 8.82 cC	70.00 ± 10.00 cB	83.33 ± 12.02 cA
	6.3	83.33 ± 3.33 bC	86.67 ± 8.82 bB	100 ± 0.00 aA	66.67 ± 13.33 bC	76.67 ± 3.33 bB	90.00 ± 5.77 bA
	12.5	96.67 ± 3.33 aC	100.0 ± 0.00 aB	100 ± 0.00 aA	80.00 ± 11.55 aC	86.67 ± 8.82 aB	100 ± 0.00 aA
*Araucaria heterophylla*	0	0.00 ± 0.0 fC	3.33 ± 3.33 fB	6.67 ± 3.33 fA	0.00 ± 0.0 fC	3.33 ± 3.33 fB	6.67 ± 3.33 eA
	0.8	10.00 ± 5.77 eC	26.67 ± 6.67 eB	56.67 ± 6.67 eA	23.33 ± 3.33 eC	50.00 ± 5.77 eB	76.67 ± 6.67 dA
	1.6	20.00 ± 5.77 dC	53.33 ± 3.33 dB	73.33 ± 8.82 dA	46.67 ± 3.33 dC	76.67 ± 3.33 dB	86.67 ± 3.33 cA
	3.1	40.00 ± 5.77 cC	63.33 ± 3.33 cB	76.67 ± 3.33 cA	66.67 ± 8.82 cC	86.67 ± 6.67 cB	93.33 ± 6.67 bA
	6.3	50.00 ± 5.77 bC	70.00 ± 5.77 bB	93.33 ± 6.67 bA	86.67 ± 3.33 bC	93.33 ± 6.67 bB	100.0 ± 0.00 aA
	12.5	70.00 ± 5.77 aC	83.33 ± 12.02 aB	100.0 ± 0.00 aA	96.67 ± 3.33 aC	100.0 ± 0.00 aB	100.0 ± 0.00 aA

* letters refer to significant difference; a–f: There is no significant difference (*p* > 0.05) between any two means, within the same column they have the same superscript letter; A, B & C: There is no significant difference (*p* > 0.05) between any two means for the same solvent, within the same row they have the same superscript letter. Three replicates were used for each concentration and ten numbers of adult pests per replicate were used.

**Table 8 plants-11-01682-t008:** Lethal concentrations of plant extracts of *Commiphora molmol* and *Araucaria heterophylla* against *Hippobosca maculata*.

Days	Plant Extracts	Solvents	LC_50_ (95%CL) *	LC_90_ (95%CL)	LC_95_ (95%CL)	Equation **	X^2^
**1**	*Commiphora molmol*	Methanol	2.31 (1.96–2.70)	11.24 (8.64–16.10)	17.60 (12.76–27.57)	1.866 ± 0.171	2.861
		Hexane	3.84 (3.27–4.56)	21.60 (15.68–33.70)	35.23 (23.91–60.73)	1.714 ± 0.156	0.733
	*Araucaria heterophylla*	Methanol	5.58 (4.60–7.02)	41.19 (26.40–79.38)	72.59 (42.51–160.78)	1.476 ± 0.153	1.476
		Hexane	1.80 (1.53–2.09)	7.47 (6.06–9.84)	11.18 (8.65–15.79)	2.079 ± 0.178	0.395
**3**	*Commiphora molmol*	Methanol	1.40 (1.15–1.64)	6.19 (5.01–8.20)	9.43 (7.25–13.53)	1.988 ± 0.183	5.842
		Hexane	1.72 (1.34–2.12)	15.08 (10.57–25.46)	27.86 (17.73–55.12)	1.326 ± 0.151	3.343
	*Araucaria heterophylla*	Methanol	2.07 (1.61–2.58)	22.23 (14.49–42.83)	43.53 (25.30–101.41)	1.245 ± 0.146	5.163
		Hexane	0.77 (0.56–0.97)	4.01 (3.31–5.11)	6.40 (5.03–8.89)	1.794 ± 0.181	2.148
**7**	*Commiphora molmol*	Methanol	0.67 (0.48–0.83)	2.51 (2.09–3.20)	3.65 (2.90–5.11)	2.2.33 ± 0.276	3.968
		Hexane	0.68 (0.45–0.91)	4.70 (36.7–6.66)	8.12 (5.87–13.28)	1.533 ± 0.187	5.665
	*Araucaria heterophylla*	Methanol	0.78 (0.18–1.05)	4.91 (3.56–18.20)	8.28 (6.51–51.32)	1.602 ± 0.187	7.916
		Hexane	0.32 (0.16–0.49)	2.06 (1.64–2.61)	3.49 (2.74–4.87)	1.586 ± 0.214	1.033

* LC_50_, _60_, and _95_ values = lethal concentration that kills 50, 90, and 95% of the exposed ectoparasite; (95%CL) = lower and upper confidence limit; ** Regression line equation; X^2^ = chi-square; Significant at *p* < 0.05 level.

**Table 9 plants-11-01682-t009:** The major chemical constituents of *Commiphora molmol* methanol extracts.

No.	M. F. *	Chemical Name (99.98%)	Area (%)	RT	Nature of Compound
1	C_15_H_24_	Cyclohexene, 4-ethenyl-4-methyl-3-(1-methylethenyl)-1-(1-methylethyl)-, (3R-trans)-	0.74	9.38	phenol
2	C_15_H_24_	(-)-á-Bourbonene	3.78	10.33	fatty acid esters
3	C_15_H_24_	Tricyclo [2.2.1.0(2,6)]heptane, 1,7-dimethyl-7-(4-methyl-3-pentenyl)-, (-)-	1.86	11.11	carboxylic acid
4	C_15_H_24_	ç-Elemene	2.11	11.40	fatty acid esters
5	C_15_H_24_	1,6-CYCLODECADIENE, 1-METHYL-5-METHYLENE-8-(1-METHYLETHYL)-, [S-(E,E)]-	1.78	12.34	fatty acid esters
6	C_15_H_24_	Aromandendrene	0.65	12.43	fatty acid ester
7	C_15_H_24_	Azulene, 1,2,3,3a,4,5,6,7-octahydro-1,4-dimethyl-7-(1-methylethenyl)-, [1R-(1à,3aá,4à,7á)]-	0.06	12.61	terpenoids
8	C_15_H_20_O	Benzofuran, 6-ethenyl-4,5,6,7-tetrahydro-3,6-dimethyl-5-isopropenyl-, trans-	0.81	12.82	heterocyclic
9	C_15_H_20_O	Benzofuran, 6-ethenyl-4,5,6,7-tetrahydro-3,6-dimethyl-5-isopropenyl-, trans-	15.35	13.17	heterocyclic
10	C_15_H_24_	ç-Muurolene	0.24	13.27	sesquiterpene
11	C_15_H_24_	Naphthalene, 1,2,3,5,6,8a-hexahydro-4,7-dimethyl-1-(1-methylethyl)-, (1S-cis)-	0.32	13.36	sesquiterpene
12	C_15_H_24_	á-Longipinene	0.20	13.51	sesquiterpene
13	C_15_H_24_	Azulene, 1,2,3,3a,4,5,6,7-octahydro-1,4-dimethyl-7-(1-methylethenyl)-, [1R-(1à,3aá,4à,7á)]-	0.08	13.61	sesquiterpene
14	C_15_H_24_	1,5-Cyclodecadiene, 1,5-dimethyl-8-(1-methylethylidene)-, (E,E)-	2.04	13.92	sesquiterpene
15	C_15_H_18_O	3,5,8a-Trimethyl-4,6,8a,9-tetrahydronaphtho [2,3-b]furan	0.99	14.36	phenol
16	C_15_H_18_O	Azulen-2-ol, 1,4-dimethyl-7-(1-methylethyl)-	0.47	15.37	acetic acid
17	C_15_H_18_O	1-NAPHTHALENOL, 4,7-DIMETHYL-2-(1-METHYLETHYL)-	30.80	15.87	phenol
18	C_15_H_18_O	(4aS,8aS)-3,8a-Dimethyl-5-methylene-4,4a,5,6,8a,9-hexahydronaphtho [2,3-b]furan	7.98	15.95	phenol
19	C_15_H_20_O	Benzofuran, 6-ethenyl-4,5,6,7-tetrahydro-3,6-dimethyl-5-isopropenyl-, trans-	0.89	16.16	heterocyclic
20	C_17_H_28_O_2_	Cyclohexanemethanol, 4-ethenyl-à,à,4-trimethyl-3-(1-methylethenyl)-, acetate, [1R-(1à,3à,4á)]-	2.37	16.35	sesquiterpene
21	C_16_H_22_O_2_	(R,5E,9E)-8-Methoxy-3,6,10-trimethyl-4,7,8,11-tetrahydrocyclodeca[b]furan	12.72	17.25	sesquiterpene lactones
22	C_15_H_24_	AZULENE, 1,2,3,4,5,6,7,8-OCTAHYDRO-1,4-DIMETHYL-7-(1-METHYLETHYLIDENE)-, (1S-CIS)-	0.29	18.15	sesquiterpene
23	C_17_H_24_O_4_	Acetic acid, 6-(1-hydroxymethyl-vinyl)-4,8a-dimethyl-3-oxo-1,2,3,5,6,7,8,8a-octahydronaphthalen-2-yl ester	2.03	19.08	phenol
24	C_15_H_20_O_3_	Reynosin	0.11	19.24	fatty acid esters
25	C_23_H_34_O_2_	Methyl 4,7,10,13,16,19-docosahexaenoate	0.18	19.32	steroids
26	C_17_H_24_O_4_	6-[1-(HYDROXYMETHYL)VINYL]-4,8A-DIMETHYL-3-OXO-1,2,3,5,6,7,8,8A-OCTAHYDRO-2-NAPHTHALENYL ACETATE	10.35	20.16	fatty acid esters
27	C_15_H_20_O_2_	FUROSARDONIN A	0.42	20.36	fatty acid esters
28	C_15_H_22_O_3_	5,8-Dihydroxy-4a-methyl-4,4a,4b,5,6,7,8,8a,9,10-decahydro-2(3H)-phenanthrenone	0.36	21.82	fatty acid esters

* Molecular formula.

**Table 10 plants-11-01682-t010:** The major chemical constituents of *Commiphora molmol* hexane extracts.

No.	M. F.	Chemical Name (100%)	Area (%)	RT	Nature of Compound
1	C_15_H_24_	Cyclohexene, 4-ethenyl-4-methyl-3-(1-methylethenyl)-1-(1-methylethyl)-, (3R-trans)-	0.88	9.36	phenol
2	C_10_H_12_O_2_	PHENOL, 2-METHOXY-4-(2-PROPENYL)-	1.65	10.30	fatty acid esters
3	C_15_H_24_	CYCLOHEXANE, 1-ETHENYL-1-METHYL-2,4-BIS(1-METHYLETHENYL)-, [1S-(1à,2á,4á)]-	4.53	10.59	carboxylic acid
4	C_15_H_24_	Tricyclo [2.2.1.0(2,6)]heptane, 1,7-dimethyl-7-(4-methyl-3-pentenyl)-, (-)-	1.51	11.08	fatty acid esters
5	C_15_H_24_	ç-Elemene	1.32	11.42	fatty acid esters
6	C_15_H_24_	1,6-CYCLODECADIENE, 1-METHYL-5-METHYLENE-8-(1-METHYLETHYL)-, [S-(E,E)]-	1.72	12.32	sesquiterpene
7	C_15_H_24_	Aromandendrene	0.65	12.43	fatty acid ester
8	C_15_H_24_	Azulene, 1,2,3,3a,4,5,6,7-octahydro-1,4-dimethyl-7-(1-methylethenyl)-, [1R-(1à,3aá,4à,7á)]-	0.57	12.41	sesquiterpene
9	C_15_H_20_O	5-ISOPROPENYL-3,6-DIMETHYL-6-VINYL-4,5,6,7-TETRAHYDRO-1-BENZOFURAN #	1.10	12.82	sesquiterpene
10	C_15_H_20_O	Benzofuran, 6-ethenyl-4,5,6,7-tetrahydro-3,6-dimethyl-5-isopropenyl-, trans-	12.09	13.07	heterocyclic
11	C_15_H_24_	ç-Muurolene	0.38	13.17	sesquiterpene
12	C_15_H_24_	GERMACRENE B	2.39	13.87	sesquiterpene
13	C_15_H_18_O	3,5,8a-Trimethyl-4,6,8a,9-tetrahydronaphtho [2,3-b]furan	1.25	14.33	sesquiterpene
14	C_15_H_18_O	NAPHTHALENE, 4-METHOXY-1,2,6,8-TETRAMETHYL-	31.98	15.76	phenol
15	C_15_H_18_O	(4aS,8aS)-3,8a-Dimethyl-5-methylene-4,4a,5,6,8a,9-hexahydronaphtho [2,3-b]furan	8.15	15.84	phenol
16	C_15_H_20_O	Benzofuran, 6-ethenyl-4,5,6,7-tetrahydro-3,6-dimethyl-5-isopropenyl-, trans-	0.82	16.05	heterocyclic
17	C_15_H_24_	(5E)-3,6,10-TRIMETHYL-4,7,8,11-TETRAHYDROCYCLODECA[B]FURAN	7.94	16.62	sesquiterpene
18	C_16_H_12_O_7_	METHYL 3-(CIS-3’-HYDROXY-5’-OXOTETRAFURAN-2’-YL)-1,4-DIOXO-1,4-DIHYDRONAPHTHLENE-2-CARNOXYLATE	8.39	17.15	phenol
19	C_15_H_24_	AZULENE, 1,2,3,4,5,6,7,8-OCTAHYDRO-1,4-DIMETHYL-7-(1-METHYLETHYLIDENE)-, (1S-CIS)-	0.23	18.10	sesquiterpene
20	C_17_H_24_O_4_	Acetic acid, 6-(1-hydroxymethyl-vinyl)-4,8a-dimethyl-3-oxo-1,2,3,5,6,7,8,8a-octahydronaphthalen-2-yl ester	10.88	20.09	phenol
21	C_13_H_14_N^2^O	2-PYRAZOLIN-5-ONE, 4-ISOPROPYLIDENE-3-METHYL-1-PHENYL-	0.98	20.71	phenol
22	C_15_H_22_O_3_	5,8-Dihydroxy-4a-methyl-4,4a,4b,5,6,7,8,8a,9,10-decahydro-2(3H)-phenanthrenone	0.59	21.78	sesquiterpene

**Table 11 plants-11-01682-t011:** The major chemical constituents of *Araucaria heterophylla* methanol extracts.

No.	M. F.	Chemical Name (100%)	Area (%)	RT	Nature of Compound
1	C_4_H_9_NO_2_S	DL-Homocysteine	0.86	4.65	terpenoid
2	C_10_H_16_	1,4-CYCLOHEXADIENE, 1-METHYL-4-(1-METHYLETHYL)-	1.22	6.68	phenol
3	C_10_H_16_	(1R)-2,6,6-Trimethylbicyclo [3.1.1]hept-2-ene	4.38	8.74	phenol
4	C_10_H_16_	à-Pinene	3.24	9.40	monoterpene
5	C_10_H_16_	1,3,7-OCTATRIENE, 3,7-DIMETHYL-	1.44	10.09	fatty acid
6	C_10_H_16_	à-Pinene	0.66	10.19	monoterpene
7	C_10_H_16_	BICYCLO [3.1.0]HEXANE, 4-METHYLENE-1-(1-METHYLETHYL)-	1.99	11.04	monoterpene
8	C_10_H_16_	CYCLOHEXENE, 1-METHYL-4-(1-METHYLETHENYL)-	12.95	13.07	monoterpene
9	C_10_H_16_O	cis-p-mentha-1(7),8-dien-2-ol	1.85	16.67	monoterpene ketone
10	C_10_H_16_O	à-Campholenal	1.29	17.70	monoterpene
11	C_10_H_16_O	Isopinocarveol	2.38	18.19	monoterpene
12	C_10_H_16_O	cis-Verbenol	3.8	18.49	monoterpene
13	C_10_H_16_O	Isopinocarveol	0.68	19.57	monoterpene
14	C_10_H_16_O	Bicyclo [3.1.1]hept-2-ene-2-methanol, 6,6-dimethyl-	0.97	20.77	terpene alcohols
15	C_13_H_26_	6-Tridecene, (Z)-	9.34	20.90	fatty acid
16	C_10_H_16_O	(-)-MYRTENOL	1.33	21.52	glycosides
17	C_10_H_14_O	Bicyclo [3.1.1]hept-3-en-2-one, 4,6,6-trimethyl-, (1S)-	2.24	22.49	terpene alcohols
18	C_15_H_24_	.alfa.-Copaene	1.85	23.79	sesquiterpene
19	C_15_H_24_	Ylangene	1.13	24.79	sesquiterpene
20	C_15_H_24_	Copaene	7.96	25.41	sesquiterpene
21	C_15_H_24_	(-)-á-Bourbonene	3.41	25.58	sesquiterpene
22	C_15_H_24_	Caryophyllene	2.63	27.22	sesquiterpene
23	C_15_H_24_	á-ylangene	1.22	28.05	sesquiterpene
24	C_15_H_24_	á-copaene	0.84	28.48	sesquiterpene
25	C_15_H_24_	ç-Muurolene	3.48	29.64	sesquiterpene
26	C_15_H_24_	Germacrene D	2.34	2984	sesquiterpene
27	C_15_H_24_	ç-Muurolene	2.07	30.20	sesquiterpene
28	C_15_H_24_	Naphthalene, 1,2,3,5,6,8a-hexahydro-4,7-dimethyl-1-(1-methylethyl)-, (1S-cis)-	3.82	30.85	sesquiterpene
29	C_15_H_24_	á-copaene	1.4	31.59	sesquiterpene
30	C_22_H_44_O_4_Si	OCTADECANOIC ACID, 9,10-EPOXY-18-(TRIMETHYLSILOXY)-, METHYL ESTER, CIS-	1.08	33.31	fatty acid
31	C_15_H_24_O	Caryophyllene oxide	10.39	34.01	phenol
32	C_15_H_24_O	4,12,12-TRIMETHYL-9-METHYLENE-5-OXATRICYCLO [8.2.0.0~4,6~]DODECANE	0.98	34.40	sesquiterpene
33	C_15_H_24_O	Caryophyllene oxide	1.18	35.04	sesquiterpene
34	C_15_H_26_O	1,1,4,7-TETRAMETHYLDECAHYDRO-1H-CYCLOPROPA[E]AZULEN-4-OL #	1.31	35.76	sesquiterpene
35	C_15_H_24_O	Caryophyllene oxide	1.50	36.42	phenol
36	C_18_H_34_O_3_	Oxiraneoctanoic acid, 3-octyl-, cis-	0.79	37.75	sesquiterpene

**Table 12 plants-11-01682-t012:** The major chemical constituents of *Araucaria heterophylla* hexane extracts.

No.	M. F.	Chemical Name (99.90%)	Area (%)	RT	Nature of Compound
1	C_10_H_16_	BICYCLO [3.1.1]HEPT-2-ENE, 2,6,6-TRIMETHYL-	0.84	5.22	terpenoid
2	C_10_H_16_	2,6,6-TRIMETHYLBICYCLO [3.1.1]HEPT-2-ENE	6.5	6.31	phenol
3	C_10_H_16_	D-Limonene	1.35	7.12	monoterpene
4	C_15_H_24_	5,5-Dimethylimidazolidine-2,4-dione	3.57	8.34	sesquiterpene
5	C_6_H_10_O	3-PENTEN-2-ONE, 4-METHYL-	1.62	8.54	alkene
6	C_3_H_6_DN	AZETIDINE-D1	8.28	9.41	saturated heterocyclic
7	C_7_H_14_O_2_	2-PENTANONE, 4-METHOXY-4-METHYL-	1.26	10.32	phenol
8	C_10_H_16_	CYCLOHEXENE, 1-METHYL-4-(1-METHYLETHENYL)-	4.11	12.62	monoterpene
9	C_10_H_15_NO_2_	Benzenemethanol, 4-hydroxy-à-[1-(methylamino)ethyl]-, (R*,S*)-	0.39	14.62	saturated heterocyclic
10	C_10_H_16_O	à-Campholenal	0.69	15.04	monoterpene
11	C_8_H_16_O	2-OCTANONE	0.44	17.38	organic aldehyde
12	C_10_H_16_O	Bicyclo [3.1.1]hept-3-en-2-ol, 4,6,6-trimethyl-, [1S-(1à,2á,5à)]-	2.90	21.19	terpenes
13	C_15_H_24_	ç-Elemene	0.23	23.62	terpenes
14	C_18_H_34_O_2_	9-OCTADECENOIC ACID (Z)-	14.60	25.95	terpenes
15	C_15_H_24_	BICYCLO [7.2.0]UNDEC-4-ENE, 4,11,11-TRIMETHYL-8-METHYLENE-, [1R-(1R*,4E,9S*)]-	0.22	26.72	terpenes
16	C_15_H_24_	à-Cubebene	0.31	27.23	terpenes
17	C_15_H_24_	1,4-METHANOAZULENE, DECAHYDRO-4,8,8-TRIMETHYL-9-METHYLENE-, [1S-(1à,3Aá,4à,8Aá)]-	0.49	28.09	terpenes
18	C_15_H_24_	(-)-á-Bourbonene	0.98	30.30	terpenes
19	C_15_H_24_	ç-Elemene	0.96	31.22	terpenes
20	C_18_H_36_O_2_	Hexadecanoic acid, ethyl ester	9.57	31.70	terpenes
21	C_18_H_34_O_2_	9-OCTADECENOIC ACID (Z)-	3.18	32.98	terpenes
22	C_15_H_24_O_2_	BICYCLO [4.4.0]DEC-2-EN-4-OL, 2-METHYL-9-(PROP-1-EN-3-OL-2-YL)-	1.42	33.79	terpenes
23	C_16_H_22_	4,4’-Dimethyl-2,2’-dimethylenebicyclohexyl-3,3’-diene	4.21	34.10	terpenes
24	C_15_H_24_	Aromandendrene	1.30	34.76	sesquiterpene
25	C_15_H_24_	á-Longipinene	0.45	35.01	sesquiterpene
26	C_27_H_46_O	CHOLEST-5-EN-3-OL (3á)-	19.15	35.54	fatty acid
27	C_26_H_44_O_5_	Ethyl iso-allocholate	0.11	36.79	fatty acid
28	C_18_H_34_O_2_	9-OCTADECENOIC ACID (Z)-	5.19	37.2	fatty acid
29	C_16_H_32_O_2_	n-Hexadecanoic acid	2.13	37.63	fatty acid
30	C_19_H_26_O_6_	ISOCHIAPIN B	3.45	40.03	fatty acid

## Data Availability

Not applicable.

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
