# Peer review of "Novel Pesticidal Efficacy of Araucaria heterophylla and Commiphora molmol Extracts against Camel and Cattle Blood-Sucking Ectoparasites"

_plants, 2022, doi:10.3390/plants11131682_

Round 1

Reviewer 1 Report

The paper title “Novel Pesticidal Efficacy of Araucaria heterophylla and Commiphora molmol Extracts against Camel and Cattle Blood-Sucking Ectoparasites” by Mohamed M. Baz et al present interesting data.

I continue to think that Analysis by GC based only in the peak are is not appropriate. The author can use only GC-MS as qualitative analysis even the identification of the getting peak was done by comparison of their mass spectra with those of mass spectral databases of GC-MS system. We can have more than one compound in the same region and with similar mass spectra. I hope the authors carefully confirmed by assessment of Kovats retention index and by MS fragmentation pattern.

However, if the other reviewers and the editor think proper, I can accept but the author must explain that is a relative quantification.

Some additional comments:

Line 30, 84, 157, … – corrcet the name “Hi. Maculata” sometimes Ha. Maculata, etc.

check in all document - scientific names must be in italic. Also, in Figure 1

line 287 – the authors use methanol or hexane or a mix of them or a sequential extraction? Must be well explained.

Line 301 define what RH is

Table 1 footnote – in yellow correct the word “ concnetratin”

Tableas 1-9, decide if it was used the simplified scientific names or not, and be consistent in all tables and manuscript ex: Commiphora molmol or C. molmol ?

Be consist, you use GC–MS, GCMS and GC/MS in the manuscript

Author Response

Reply to Review report 1

Thanks for the reviewer one for providing carful revision and helpful comments of the manuscript

Point#: I continue to think that Analysis by GC based only in the peak are is not appropriate. The author can use only GC-MS as qualitative analysis even the identification of the getting peak was done by comparison of their mass spectra with those of mass spectral databases of GC-MS system. We can have more than one compound in the same region and with similar mass spectra. I hope the authors carefully confirmed by assessment of Kovats retention index and by MS fragmentation pattern.

However, if the other reviewers and the editor think proper, I can accept but the author must explain that is a relative quantification.

Response#: Author has been using GC-MS as qualitative analysis to report all components of the injected extract, the identification of the getting peak was done by comparison of their mass spectra with those of mass spectral databases of GCMS system with last update versions of NIST and Wiley database (Ashmawy et al. 2018; El-Hefny et al. 2018) and a lot of authors. later, we will use the assessment of Kovats retention index and by MS fragmentation pattern.

Some additional comments:

Point#: Line 30, 84, 157, … – corrcet the name “Hi. Maculata” sometimes Ha. Maculata, etc.

Response#: Corrected throughout the text as Hi. maculata

Point#: Check in all document - scientific names must be in italic. Also, in Figure 1.

Response#: Done.

Point#: line 287 – the authors use methanol or hexane or a mix of them or a sequential extraction? Must be well explained.

Response#: Each extract was evaluated individually, explained in the text.

Point#: Line 301 define what RH is

Response#: Relative humidity, explained in the text.

Point#: Table 1 footnote – in yellow correct the word “ concnetratin”

Response#: Done.

Point#: Table as 1-9, decide if it was used the simplified scientific names or not, and be consistent in all tables and manuscript ex: Commiphora molmol or C. molmol ?

Response#: Corrected.

 Point#: Be consist, you use GC–MS, GCMS and GC/MS in the manuscript

Response#: GC–MS was used.

Reviewer 2 Report

Review of MS plants-178014: Pesticidal Efficacy of Araucaria heterophylla and Commiphora molmol Extracts against Camel and Cattle Blood-Sucking Ectoparasites.

Overall, the authors improved the MS. In particular, several sections of it are now much more fluid and understandable. I advise authors to double check the text in the different sections and verify the different typing errors present.

Specifically, further corrections are needed.

Line 295-296. How many individuals were used in the preliminary test? Specify for each replica. Also I cannot understand what “Treated envelopes [44] were used” is.

Were the adults dipped in the different dilutions?

Specify in this section whether each test included a set control groups;

Line 321. lethal concentration (LC): (LC50) or (LC50 and LC90.

In the caption of the table the authors wrote “Three replicates were used for each concnetratin and 10 pests per repicate were used”. Correct the English sentence. Which means 10 pests better to write "ten numbers of adults ..."

Author Response

Review report 2

The authors would like to thank Reviewer 2 for careful revision and helpful comments to the manuscript.

Point#: Line 295-296. How many individuals were used in the preliminary test? Specify for each replica. Also I cannot understand what “Treated envelopes [44] were used” is.

Response#: Preliminary test (each contained 30 adult pests, grouped in three replicates). More details about the treated envelops were added to the text.

Point#: Were the adults dipped in the different dilutions?

Response#: No, more details about the treated envelops were added to the text.

Point#: Specify in this section whether each test included a set control groups;

Response#: yes, we set a control groups with each test.

Point#: Line 321. Lethal concentration (LC): (LC50) or (LC50 and LC90.

Response#: Such information was explained in the aims of the work.

"The study's main goals were to investigate the novel pesticidal effect of methanol and hexane extracts of myrrh and Polynesian pine against four camel and cattle blood-sucking ectoparasites, calculate their lethal concentration values that kills 50, 90, and 95% of the exposed ectoparasites (LC50, 60, and 95, respectively), and investigated their phytochemical analyses."

Point#: In the caption of the table the authors wrote “Three replicates were used for each concnetratin and 10 pests per repicate were used”. Correct the English sentence. Which means 10 pests better to write "ten numbers of adults ..."

Response#: Corrected as recommended.

Reviewer 3 Report

 My comments are as under:

  • The abstract must include data regarding the critical finds by the authors in terms of data of important findings. 
  • The introduction must have a clear hypothesis and significantly develop the second paragraph of this manuscript.
  • Overall there is the repetition of the information which could be avoided.  
  •  Check figure ligands; they are carelessly written.
  • Discussion should include more information and references related to the relevant and related works. 
  • Restructure and carefully edit the conclusion section.

Author Response

Review report 3

The authors appreciated reviewer 3 for his/her helpful comments and Suggestions

Point#: The abstract must include data regarding the critical finds by the authors in terms of data of important findings. 

Response#: 200 words only were allowed for the abstract and it covers the most important findings.

Point#: The introduction must have a clear hypothesis and significantly develop the second paragraph of this manuscript.

Response#: Corrected as recommended.

Point#: Overall there is the repetition of the information which could be avoided.  

Response#: The manuscript was revised and corrected as recommended.

Point#: Check figure ligands; they are carelessly written.

Response#: Figure ligands were corrected

Point#: Discussion should include more information and references related to the relevant and related works. 

Response#: Corrected as recommended and more information and relevant references were added.

 Point#: Restructure and carefully edit the conclusion section.

Response#: Conclusion was revised, and more information was added as suggested.

Round 2

Reviewer 3 Report

Authors have significantly improved their manuscript; therefore, it deserves to be published.

Author Response

Review report 3

The authors appreciated reviewer 3 for his/her helpful comments and Suggestions

Point#: English language and style are fine/minor spell check required. 

Response#: English language and style were checked.

This manuscript is a resubmission of an earlier submission. The following is a list of the peer review reports and author responses from that submission.

Round 1

Reviewer 1 Report

The paper title “Novel Pesticidal Efficacy of Araucaria heterophylla and Commiphora molmol Extracts against Camel and Cattle Blood-Sucking Ectoparasites” by Mohamed M. Baz, Hanem F. Khater, Rowida S. Baeshen, Abdelfattah Selim, Yasser A. El-Sayed and Maysa M. Hegazy present interesting data but not well discussed or presented. The chromatography analyses present some weaknesses, so my opinion is rejecting the paper.

Abstract: the sentence “Botanicals effectively control pests” must be deleted or completed

Keywords: scientific names must be in italic

The Introduction must be increased.

According the guidelines for author the Materials and Methods section must be after the results and discussion. Please correct this point

2.2. Plant materials collection: it will be possible include a figure of the Araucaria heterophylla

Please revise this sentence “….were  prepared with the addition of 1 ml of tween 80 as an emulsifier to ensure the complete  solubility of the extract in water”

Indicate the type of solvents used

It was important also mentioned how many replications of each experiment was made

In material and methods section miss also how the LC was calculated

In 2.4. Bioassays, added more information about how the application was made and, in each concentrate, and if there simulate exactly the application in the animals.

In material and methods section a subsection about the data analysis is need as well the software used.

The first sentence of the 3.1. Pesticidal effects section, from line to 108 to line 113, must be in material and methods.

Table 2 – each table must be self-readable, so included ad footnote of the table the means of each abbreviation used [Check in all document]. In this table indicate also the mean of the numbers in brackets and put the standard deviations for each LC. Wat means the column Slope ±SD?? Is not explain in any part as well the column X2. Same comments for other related tables.

The discussion of the results is very poor. Could be increased.

Analysis by GC based only in the peak are is not appropriate. Why the authors did not use an internal standard and quantified based on its concentration? How the author knows if there is the compound identified when sometimes we have more than one with the same percentage of similarities and quite same Retention time. This analysis is not very scientific for the quality of Plants Journal.

Fig 4 is impossible to understand and in Fig 3 there are some coeluted peak that corroborate my previously comment

Reviewer 2 Report

Review of MS plants-1737533: Pesticidal Efficacy of Araucaria heterophylla and Commiphora molmol extracts against Camel and Cattle Blood-Sucking Ectoparasites.

The MS addresses an interesting topic concerning the control of ectoparasites affecting livestock in general and capable of transmit infectious agents to vertebrate hosts, thus causing major constraints to livestock health. In consideration of the need to propose natural extracts for the containment of these species, the topic discussed presents interesting results. The extension of use both to ectoparasitic insects (Haematopinus eurysternus; Hippobosca maculate) and to bloodsucking ticks (Hyalomma dromedarii, Rhipicephalus) lays the foundations for further studies.

It should be noted, however, that the MS presented does not contain various information that is necessary in any scientific work. In particular, it is necessary to report in detail the number of replicas used for each species investigated and the number for each replicate. Additional information is needed for each parasite species. For example, in the tests used it is not reported whether male and female were used indifferently or whether all stages were used. LC50 tests are valid only for the species that is tested and the specific conditions used.

Was used Petri-plate bioassay procedure?

Have corrections been made to the mortality data or simple correction for natural mortality ?;

What computational method was used to derive a median lethal concentration (LC50) from concentration-mortality data produced by an acute mortality test?

Among the aspects to be taken into consideration it is necessary to verify in future applications the effects of the use of surfactants as tween 80 (see: https://doi.org/10.1016/j.etap.2014.12.015);

Provide more details on exposure and routes of botanicals used.

Specific comments

Line 56. Add latin name of botanical species;

Line 60. In this paragraph add when (also the time intervals of collection) of the specimens; The same for paragraph 2.2;

Line 69. Specify whether hardened resin extracted from plants was obtained;

Line 90. Indicate for how many days mortality was checked;

Line 109-113. This sentence seems redundant already in the materials and methods that had been indicated in the previous paragraphs;

Line 116. mortality percent (MO%);

Line 117. LC50 (50% Lethal Concentration) at first time;

Line 146. Authors means a large amounts here "There is a lake of studies about using botanicals";

Line 147-155- The authors make no reference to the repellent action of some of these species also used against other ectoparasites or parasites in general;

In table 1 it is necessary to improve the caption and in particular it must be self-explanatory. It is MO%;

In table 1 and also in others tables add the column of numbers of specimens used.

Reviewer 3 Report

This manuscript demonstrated the pesticidal activities of two plant extracts against two blood-sucking tick species. However, this manuscript lacks some important points of the manuscript such as the significance of two plant sources for pesticidal activities in the Introduction, careful experimental procedures in Materials and methods, and appropriate Discussion of experimental results. I add major and minor comments and suggestions to the manuscript file for the improvement of the mansucript.  
